# Targeting mTOR Pathway in PTEN Deleted Newly Isolated Chordoma Cell Line

**DOI:** 10.3390/jpm13030425

**Published:** 2023-02-27

**Authors:** Francesca Pagani, Magdalena Gryzik, Elena Somenza, Manuela Cominelli, Piera Balzarini, Alberto Schreiber, Davide Mattavelli, Piero Nicolai, Francesco Doglietto, Pietro Luigi Poliani

**Affiliations:** 1Pathology Unit, Department of Molecular and Translational Medicine, University of Brescia, 25123 Brescia, Italy; 2Unit of Otorhinolaryngology-Head and Neck Surgery, ASST Spedali Civili Brescia, Department of Medical and Surgical Specialties, Radiological Sciences, and Public Health, University of Brescia, 25123 Brescia, Italy; 3Section of Otorhinolaryngology-Head and Neck Surgery, Department of Neurosciences, University of Padova—Azienda Ospedale-Università di Padova, 35128 Padova, Italy; 4Fondazione Policlinico Universitario Agostino Gemelli IRCCS, Catholic University School of Medicine, 00168 Rome, Italy

**Keywords:** chordoma, CH3 cell line, mTOR, rapamycin

## Abstract

Chordomas are rare primary malignant tumours of notochordal origin usually arising along the axial skeleton with particular predilection of the skull base and sacrococcygeal region. Albeit usually slow-growing, chordomas can be aggressive mostly depending on their invasive behaviour and according to different histotypes and molecular alterations, including *TBXT* duplication and *SMARCB1* homozygous deletion. Partial or complete PTEN deficiency has also been observed. PTEN is a negative regulator of the Akt/mTOR pathway and hyperactivation of Akt/mTOR in cells lacking PTEN expression contributes to cell proliferation and invasiveness. This pathway is targeted by mTOR inhibitors and the availability of in vitro models of chordoma cells will aid in further investigating this issue. However, isolation and maintenance of chordoma cell lines are challenging and *PTEN*-deleted chordoma cell lines are exceedingly rare. Hereby, we established and characterized a novel human *PTEN*-deleted chordoma cell line (CH3) from a primary skull base chordoma. Cells exhibited morphological and molecular features of the parent tumour, including *PTEN* loss and expression of Brachyury and EMA. Moreover, we investigated the activation of the mTOR pathway and cell response to mTOR inhibitors. CH3 cells were sensitive to Rapamycin treatment suggesting that mTOR inhibitors may represent a valuable option for patients suffering from *PTEN*-deleted chordomas.

## 1. Introduction

Chordoma is a rare bone tumour with an incidence of 0.08 cases per 100,000 individuals and a median age at diagnosis of 50–60 years [1,2] arising from undifferentiated remnants of the embryonic notochord. Chordomas are mainly located in the skull base and sacrococcygeal region and comprise different histotypes, including conventional, chondroid, dedifferentiated and poorly differentiated chordomas [2,3,4]. The most common type, the conventional chordoma, is characterized by cords and nests of large epithelioid cells with clear to light eosinophilic vacuolated cytoplasm (physaliphorous cells) within an extracellular abundant myxoid matrix. Chondroid chordoma, a subtype of conventional chordoma, displays areas of cartilaginous differentiation reminiscent of chondrosarcoma. Dedifferentiated chordoma exhibits a biphasic phenotype with areas of conventional chordoma along with areas of high-grade sarcomatous features composed of spindle and pleomorphic cells. Poorly differentiated chordoma is a rare histotype arising mainly in children and young adults and composed of nests of mitotically active epithelioid cells with scattered intracytoplasmic vacuoles lacking physaliphorous cell morphology and extracellular myxoid stroma and characterized by *SMARCB1* homozygous deletion and loss of INI1 expression [3,4,5]. The latter two histotypes are considered the most aggressive histotypes. Brachyury, a transcription factor encoded by the T gene (6q27) which normally is transiently expressed in embryonic notochord and required for its development, is considered the major marker of chordomas. It is highly expressed in chordomas and has been suggested to be involved in chordomagenesis, hence it is considered a potential therapeutic target [6,7]. In addition to Brachyury, chordomas also express Epithelial Membrane Antigen (EMA), low molecular weight Cytokeratins (Cytokeratin 8, 18, and 19) and S100 protein [3]. Together with the histopathological analysis, the diagnosis is based on computed tomography and magnetic resonance imaging. Due to the nonspecific clinical symptoms, which can also vary depending on the tumour location, patients are often diagnosed at an advanced stage of the disease [2]. Chordomas are slow-growing and aggressive malignancies resistant to conventional chemo- and radiotherapy. The first-line treatment is based on surgical resection of the tumour, which together with the tumour stage at the diagnosis, is one of the major prognostic factors [2]. Generally, five-year overall age-adjusted relative survival is estimated to be approximately 61% [2], while the recurrence rate is 43–85% [4]. Thus, it is of paramount importance to develop new therapeutic strategies, which greatly depend on both histological and molecular features [8]. For this reason, different pre-clinical models of chordoma have been generated in the last few years [4]. The availability of chordoma cell lines isolated from primary tumours of different types and with a different molecular profiles represents a necessary tool to investigate chordoma cell biology and to identify new therapeutic targets. The first isolated human chordoma cell line U-CH1, derived from a recurrent sacral chordoma, has allowed characterizing the genetic aspects of this malignancy [9] and the JHC7 cell line, derived from sacral primary chordoma, to identify Brachyury as a molecular marker specific for chordoma [6]. Although the isolation and maintenance of the chordoma cell line can be challenging, up to 25 human chordoma cell lines have been established so far, differing in tumour location, disease status and molecular/genetic features [4]. However, Phosphatase and Tensin Homolog (PTEN) deleted chordoma cell lines are exceedingly rare. Thus, we aimed to isolate a novel *PTEN*-deleted chordoma cell line as a valuable pre-clinical model for investigating the functional role of *PTEN* deficiency in chordomas. The present work describes a new human chordoma cell line, named CH3, derived from a skull base primary chordoma with conventional and chondroid features. Cells were characterized for morphological and molecular features demonstrating *PTEN* loss along with a similar phenotype of the parental primary tumour, including a physaliphorous phenotype and Brachyury, Cytokeratins, EMA and S100 expression. In addition, we aimed to investigate the activation of the Mammalian target of the rapamycin (mTOR) pathway and the sensitivity to the mTOR inhibitor Rapamycin.

## 2. Materials and Methods

### 2.1. Patient and Tissue Sample

This study was conducted in compliance with the Declaration of Helsinki and with policies approved by the Ethics Board of Spedali Civili di Brescia, University of Brescia for retrospective and exclusively observational study on archival material obtained for diagnostic purposes and patient consent was not needed (Delibera del Garante n. 52 del 24/7/2008 and DL 193/2003). Histological diagnosis was revised according to the World Health Organization criteria [10] and formalin-fixed paraffin-embedded (FFPE) tissue sections were selected based on adequate tissue preservation, as assayed by Hematoxylin and Eosin (H&E) staining. Information regarding clinical features, treatment and the outcome was collected from the medical records.

### 2.2. Immunohistochemistry (IHC)

Briefly, 2-μm-thick paraffin sections were obtained from FFPE. Sections were de-waxed, re-hydrated and endogenous peroxidase activity was blocked with 0.3% H_2_O_2_ in methanol for 20 min. Antigen retrieval (when necessary) was performed using a microwave or a thermostatic bath in either 1.0 mM EDTA buffer (pH 8.0) or 1 mM Citrate buffer (pH 6.0). Sections were then washed in TBS (pH 7.4) and incubated for one hour or overnight in the specific primary antibody diluted in TBS 1% bovine serum albumin. The primary antibodies used are listed in Appendix A. The reaction was revealed by using Novolink Polymer (Leica Microsystems GmbH, Wetzlar, Germany) or Dako EnVision+Dual Link System Peroxidase (DakoCytomation, Glostrup, Denmark) followed by DAB and slides counterstained with Hematoxylin. Images were acquired with a Nikon DS-Ri2 camera (4908 × 3264 full-pixel) mounted on a Nikon Eclipse 50i microscope equipped with Nikon Plan lenses (10×/0.25; 20×/0.40; 40×/0.65; 100×/1.25) using NIS-Elements 4.3 imaging software (Nikon Corporation, Tokyo, Japan).

### 2.3. Fluorescence In Situ Hybridization (FISH)

*PTEN* FISH analysis was performed with the Vysis PTEN/CEP10 FISH Probe kit (Abbott, IL, USA) according to the manufacturer’s protocol. The LSI/PTEN probe is labelled with SpectrumOrange and hybridizes to the 10q23 region that contains the PTEN gene. The CEP10 probe is labelled with SpectrumGreen and hybridizes to alpha satellite sequences specific for chromosome 10. Finally, sections were counterstained with DAPI/Antifade Solution.

### 2.4. Isolation and Maintenance of Chordoma Cells

The chordoma tissue sample from the patient was provided according to the ethical requirements of the institutional committee on human experimentation. The study was approved by the local IRB (NP2066). Briefly, tissues were mechanically dissociated and digested with collagenase type I (Thermo Fisher Scientific, Waltham, MA, USA). The human chordoma cell line, hereinafter referred to as the CH3 line, was cultured on dishes coated with collagen (122-20, Cell Applications Inc., san Diego, CA, USA) in Iscove’s Modified Dulbecco’s Medium (Euroclone, Milan, Italy) and RPMI-1640 (Euroclone) in ratio 4:1, supplemented with 10% endotoxin-free fetal bovine serum (Euroclone), 100 U/mL/100 mg/L penicillin/streptomycin (Euroclone), MEM Non-Essential Amino Acids (Euroclone), and 20 ng/mL Human Fibroblast Growth Factor (FGF, AF-100-18B, PeproTech, Ltd, London, GB). Cells were maintained at 37 °C in a humidified atmosphere with 5% CO_2_ and cultures were checked monthly for mycoplasma infection using N-GARDE Mycoplasma PCR Reagent Set (Euroclone).

### 2.5. Cell Blocks Preparation

2 million cells were resuspended in physiological solution (NaCl 0.9%); plasma and thromboplastin were added dropwise until the formation of a clotted sphere. The sphere was immediately formalin-fixed and paraffin-embedded. Cell blocks were cut into 2 µm sections and processed for IHC as previously described.

### 2.6. Immunofluorescent Staining (IF)

The cells (2 × 10^4^ cells) were seeded on glass coverslips coated with collagen. After washing with 1X phosphate-buffered saline (PBS), cells were fixed with 4% paraformaldehyde. Cells were washed three times with PBS and permeabilized by incubation with 0.1% Triton X-100 in PBS for 5 min. After three washes, cells were stained with CellBrite Blue Cytoplasmic Membrane-Labeling Kit (30024, Biotium GmbH, Eching, Germany) or directly proceeded for the blocking step. After 2–3 washes, cells were incubated with 5% FBS in PBS for 15 min, followed by incubation with primary antibody (Appendix A) for 1 h. After washing thrice, cells were incubated with the Alexa Fluor 488-conjugated goat anti-mouse IgG (A11008, Invitrogen, Waltham, MA, USA) at 1:1000 dilution for 1 h. After washing with PBS, coverslips were mounted on glass slides without or with DAPI staining and sealed. Images were collected using a Zeiss fluorescence Axiovert microscope with a CCD black-and-white TV camera (SensiCam-PCO Computer Optics GmbH, Kelheim Bavaria, Germany) and processed by ImageJ software.

### 2.7. RNA Preparation and Quantitative Real-Time PCR (qRT-PCR)

Total cell RNA was recovered with RiboEx (301-001, GeneAll Biotechnology Co. Ltd., Seoul, Republic of Korea) according to the manufacturer’s instructions. Reverse transcription was performed using 1.5 µg RNA and iScript cDNA Synthesis Kit (Bio-Rad Laboratories, Nobel Drive Hercules CA, USA). Samples were used for quantitative reverse transcription polymerase chain reaction (qRT-PCR) assay using iTaq Universal SYBR Green Supermix (#1725124, Bio-Rad), according to the manufacturer’s instructions. The following primers were used: mTOR For 5′-GGAGGAGAAATTTGATCAGG-3′, Rev 5′-GGGCAACAAATTAAGGATTG-3′; RICTOR For 5′-AAATGCATGAAGAAGCAGAG-3′, Rev 5′-AACAGTGTACAGAAGATACTCC-3′; RAPTOR For 5′-CGGAGTTTCCTTTAACAGTG-3′, Rev 5′-CTGTTGAGTACTTTCATGGC-3′; GAPDH For 5′-ACAGTTGCCATGTAGACC-3′, Rev 5′-TTGAGCACAGGGTACTTTA-3′. The levels of mRNA are expressed as 2ˆ^(−dCt)^ related to GAPDH.

### 2.8. Cell Treatment

The cells (5 × 10^3^ cells/well) were seeded on 96-well plates coated with the collagen and treated with 0–1–10-100–250–500–750–1000 nM Rapamycin (dissolved in DMSO; R0395, Sigma-Aldrich Ltd., Darmstadt, Germany) for 10 days. Cell viability was measured by MTT assay. For IHC pS6 expression, the cells (4 × 10^5^ cells/well) were seeded on 6-well plates coated with the collagen and treated with 0–100–1000 nM Rapamycin for 10 days. After the treatment, cells were detached with trypsin and centrifuged with a cytocentrifuge at 800 rpm for the thin-layer preparation. The slides were fixed with Bio-Fix (Bioptica, Milan, Italy) and processed for IHC as previously described.

### 2.9. Cell Viability Assay (MTT)

The cells were incubated in 0.5 mg/mL MTT (3-[4,5-dimethyl-2-thiazolyl]-2,5-diphenyl-2H-tetrazolium bromide, V13154, Invitrogen) in cell medium for 3.5 h at 37 °C. The insoluble formazan was dissolved in DMSO (Sigma) and the absorbance was measured at 540 nm using an EnSight Multimode plate reader (Perkin Elmer, Waltham, MA, USA).

### 2.10. Statistical Analysis

Data are presented as mean +/− standard error of the mean (SD). Data from experiments are expressed as a percentage with respect to non-treated cells. Comparison of values between untreated and treated cells was performed by a one-way ANOVA test corrected by Dunnett’s test (GraphPad Prism6, GraphPad Software, Inc., La Jolla, CA, USA). Differences were defined as significant for *p* < 0.05.

## 3. Results

### 3.1. Case Presentation

A 45-year-old man complained of persistent nasal obstruction. Nasal endoscopy showed a grey-pink mass filling the nasopharynx and obstructing the choana of both sides. Brain computed tomography (CT) scan showed a lesion of the posterior wall of the nasopharynx eroding the lower clivus on the left side. At MRI the mass was hyperintense on T2 weighted sequences and extended to the lower clivus and prevertebral muscles. No transdural extension was evident (Figure 1). These findings suggested, on the whole, the diagnosis of chordoma.

A biopsy under endoscopic guidance confirmed the diagnosis. The patient underwent an endoscopic endonasal resection. Briefly, a nasoseptal flap was harvested on the right side. Middle turbinectomy, resection of the posterior part of the inferior turbinate, and middle antrostomy were performed bilaterally together with a posterior septectomy to create an adequate surgical corridor to the lesion. A transrostral sphenoidotomy was performed and the anterior wall of both sphenoid sinuses was completely removed. The sphenoidal floor was drilled out and the anterior foramen lacerum was exposed on both sides. Intraoperatively, a macroscopic involvement of the longus colli muscle on the left side was evident, as anticipated by the MRI. Then, the lesion was removed together with the prevertebral muscles on both sides. The mid- and lower-clivus were totally exposed and widely drilled until the healthy bone was evident; on the left side, where tumour extension into the clivus was more relevant, the inner periosteum was exposed. A gross total resection was achieved. No dural breach was caused and no CSF leak was evident. The nasoseptal flap was adapted to cover the clival bone and inner clival periosteum and bolstered with absorbable haemostatic gelatin sponges (Cutanplast^®^, Milan, Italy). Nasal cavities were packed with Lyofoam^®^ Max (Molnlycke Health Care ©, Gothenburg, Sweden). The postoperative course was uneventful. Nasal packing was removed on the second postoperative day and the patient was discharged on the 4th postoperative day. A definitive pathological examination confirmed the involvement of prevertebral muscles, while the sphenoidal rostrum, sphenoidal floor, and sphenoid mucosa were negative. Resection margins were uninvolved. Postoperative MRI confirmed no residual disease was present in the surgical bed (Figure 2a,b). The patient underwent adjuvant proton therapy 3 months after tumour resection. The treatment was well tolerated. The surgical bed healed properly and no bone exposure was evident during outpatient nasal endoscopies. No complications nor tumour recurrence were recorded during follow-up and the patient is still alive and free of disease 40 months after the end of treatments (Figure 2c,d).

### 3.2. Tumour Characterization

Grossly, we received multiple tissue fragments ranging from 5 to 15 mm with a gelatinous grayfish cut surface. Intraoperative frozen section biopsies revealed permeation of the sphenoid bone mainly involving the clivus and infiltration of the adjacent tissues with extension to the left lateral wall of the rhinopharynx and prevertebral muscles. Detailed histopathological examination on formalin-fixed paraffin-embedded (FFPE) tissue specimens highlighted features of conventional chordoma composed of large epithelioid cells with clear to light eosinophilic or bubbly vacuolated cytoplasm (physaliphorous cells), arranged in cords or small nests and embedded within an extracellular myxoid matrix. Tumour cells exhibited a moderate degree of nuclear atypia or pleomorphism and low mitotic activity. Chondroid differentiation was focally observed. Histopathological features concerning dedifferentiation, such as increased cellularity, subtle spindling, mitosis and necrosis were absent (Figure 3a). Tumour cells were diffusely immunoreactive for EMA, cytokeratin, S100 and Brachyury (Figure 3b). The proliferation index was heterogeneously distributed ranging from 2–3% and up to 7–8% in the more densely cellular areas and SMARCB1 (INI1) expression was retained. Interestingly, PTEN expression was absent or barely detected according to the heterozygous deletion of the PTEN gene as assayed by FISH analysis (Figure 3c).

### 3.3. Cell Line Establishment and Characterization

The chondroid chordoma tissue obtained from the surgical resection was processed by mechanical dissociation and enzymatic digestion. The cell suspension was seeded on the collagen-coated dish in a complete growth medium (as described in the Materials and Methods Section). Accordingly, the low proliferation index of the tumour exhibited a slow growth rate. The analysis of the cell proliferation curve in passages 2 to 5 showed that the cells benefited from the addition of FGF (data not shown). This allowed us to reach the Population Doubling Level (PDL) of about 50 at passage 30. After the cell line establishment, the cells were characterized in terms of the chordoma phenotype. Microscopic analysis showed a physaliphorous cell morphology, typical for the conventional chordoma, with abundantly vacuolated cytoplasm (Figure 4, panel a and inset). Immunohistochemistry on cell block (cells at passage 20) and immunofluorescence on cells (passages 10 and 33) for Brachyury and EMA show that the cell culture maintains the phenotypical profile of the primary tumour during passages (Figure 4b). The FISH analysis confirmed *PTEN* loss and immunohistochemical staining on the cell block showed loss of PTEN expression (Figure 4c), in line with the primary tumour. Results indicate that the human chordoma CH3 cell line has been successfully isolated and established with preserved features of the original tumour.

### 3.4. mTOR Pathway

mTOR consists of two major complexes, mTOR complex 1 (mTORC1) and 2 (mTORC2), and is involved in multiple signalling pathways known to be activated in many types of tumours [11]. Several studies have shown that the mTOR pathway is hyperactivated in chordomas and have suggested it as a possible therapeutic target [12,13,14,15]. It is well known that PTEN loss results in the constitutive activation of the Akt/mTORC1 signalling pathway and contributes to the development of sporadic chordomas [16,17]. Thus, mTORC1 inhibition may provide clinical benefits to chordoma patients. To further investigate this issue, we analysed the activation state of the mTOR pathway in both the primary tumour and CH3 chordoma cell line. In the tumour sample the phosphorylated forms of S6 ribosomal protein (pS6) and N-myc downstream-regulated gene 1 (pNDRG1), which are markers of mTORC1 and mTORC2 respectively, were highly expressed (Figure 5a). Analysis of the CH3 cells confirmed the activation of the mTOR pathway at both mRNA and protein levels. mRNA expression levels of mTOR, Raptor (mTORC1) and Rictor (mTORC2) have been detected in CH3 cells and were maintained during passages (Figure 5b). IHC analysis on the FFPE cell block showed expression of both pS6 and pNDRG1, although showing lower intensity as compared to the primary tumour, particularly for pNDRG1 (Figure 5c). We next treated CH3 cells with different concentrations of Rapamycin, a robust mTOR inhibitor targeting mTOR1, for 10 days and evaluated the cell viability. Treatment had a significant effect already at the Rapamycin dose of 10 nM decreasing the cell viability by 13% (Figure 5d). However, the increase of the doses up to 1000 nM did not change drastically the cell viability that was maintained at 75–80%. The analysis of the pS6 expression in Rapamycin-treated cells cytospin confirmed the effectiveness of the treatment demonstrating that the decrease in the cell viability was due to the deployment of mTOR-positive cells (Figure 5e). Overall, data indicate that the primary chordoma tumour and, to a lesser extent, its derived CH3 cells, exhibit mTOR hyperactivation that may be targeted by a specific pharmacological treatment.

## 4. Discussion

Chordomas are aggressive slow growing tumours arising within the axial skeleton and diffusely infiltrating the surrounding bone and soft tissues. As a result, chordomas are poorly responsive to conventional cytotoxic chemotherapy [16,18] and radical surgical resection is difficult to obtain with a high rate of recurrence [19]. Hence, the investigation of more valuable treatment options represents a critical need. The pathogenic mechanisms of chordomas remain unclear. However, emerging evidence indicates that the deregulation of signaling pathways involving PI3K, Akt, PTEN, TSC1 or TSC2 may play a crucial role in chordomagenesis through the aberrant activation of the mTOR pathway [13]. mTOR is involved in the regulation of numerous cellular functions and mTOR hyperactivation is frequently observed in various types of cancers [20] contributing to cell proliferation, tumour initiation and progression [21,22,23]. Interestingly, chordomas have been reported in patients with tuberous sclerosis complex (TSC), a multisystem syndrome due to TSC1 or TSC2 alterations resulting in constitutive mTOR hyperactivation, suggesting an etiological role of TSC gene alterations in chordomagenesis [24,25]. We have previously reported that in an experimental model of TSC, concurrent hyperactivation of mTORC1 and Akt pathways mediated by co-deletion of TSC1 and PTEN is required for the formation of subependymal cell astrocytoma, a TSC-associated tumour [26]. Indeed, PTEN molecular alterations are frequently found in different tumours [27] and PTEN loss has also been reported in chordomas, recognized as a molecular alteration promoting cell proliferation and invasiveness [16,17,28]. As such, the inactivation of PTEN may be responsible for mTORC hyperactivation resulting in the development of at least a subset of sporadic chordomas. PTEN is an important negative regulator of the Akt/mTOR pathway with consequent mTORC1 hyperactivation in cells lacking PTEN resulting in constitutive phosphorylation of S6K, S6 and 4EBP1, which is potently inhibited by mTOR inhibitors. Overall, these observations suggest that treatment with mTOR inhibitors may provide clinical benefits to chordoma patients. mTOR inhibitors, including rapamycin, have been approved for a wide range of tumours [29,30] and they have been also proposed for chordoma treatment [14,31,32]. In this study, we describe a skull base chordoma harbouring *PTEN* deletion with consequent loss of PTEN expression and activation of the mTOR pathway. In addition, we have derived from this tumour a cell line that preserved in vitro the immunophenotypical and molecular features of the parent tumour, such as expression of Brachyury and EMA, as well as the physaliphorous aspect, typical for conventional chordoma as established by WHO [33]. Due to the low proliferation index (7–8%), the establishment of the cell line was challenging, requiring the optimization of cell culture conditions, allowing it to reach a PDL of about 50 at passage 30. *PTEN*-deleted chordomas are rare and, as far as we know, chordoma cell lines with *PTEN* loss derived from primary skull base chordomas have not been described so far. Moreover, we provide evidence that our cell line exhibits mTOR activation and Rapamycin treatment significantly decreases cell viability by targeting selectively the mTOR-positive cells. Of note, it was reported that PTEN-negative cells have enhanced sensitivity to mTOR inhibitors [34,35], indicating that Rapamycin treatment in PTEN mutated tumours may be even more effective as compared to sporadic chordomas with a preserved expression of PTEN providing a rationale for the treatment with mTOR inhibitors and making our cell line a unique and useful preclinical model. Given the low incidence of chordomas, further investigations will be necessary to confirm PTEN inactivation in a larger cohort of sporadic chordomas. However, the availability of PTEN-deleted cell lines represents a useful tool to further investigate the functional role of PTEN deficiency and Akt/mTORC1 hyperactivation associated with chordomas and prove if mTOR inhibition is effective in suppressing tumour growth.

## Figures and Tables

**Figure 1 jpm-13-00425-f001:**
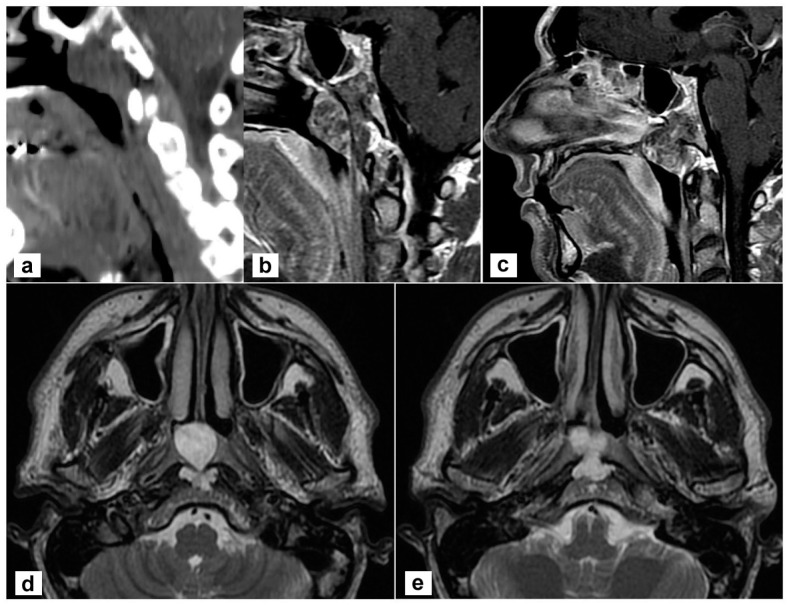
Radiological findings of the primary tumour. (**a**,**b**) Sagittal view on a paramedian plane on the left side showing the erosion of the upper portion of the lower clivus (a, computed tomography; b, T1-weighted MR sequence with contrast agent); (**c**) The lesion originates from the clivus and abutt to the clivus obstructing both choane (T1-weighted sequence with contrast); (**d**,**e**) the lesion is hyperintense in T2-weighted sequence; the involvement of prevertebral muscles is evident.

**Figure 2 jpm-13-00425-f002:**
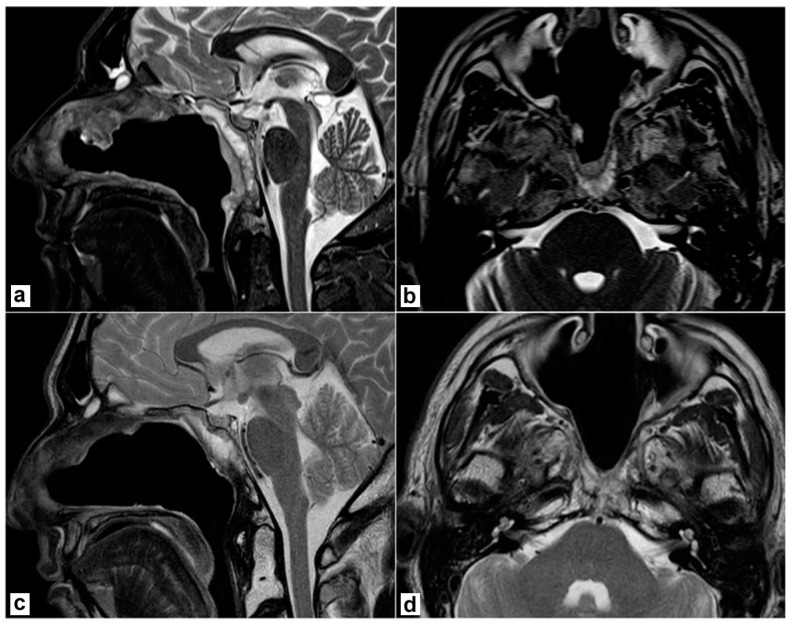
Postoperative magnetic resonance imaging (T2 weighted sequences). (**a**,**b**) 2 months after surgery, before proton therapy. No residual disease is detectable; the nasoseptal flap is vital. (**c**,**d**) 2 years after surgery. No recurrence is detectable; the post-radiotherapy scar is evident.

**Figure 3 jpm-13-00425-f003:**
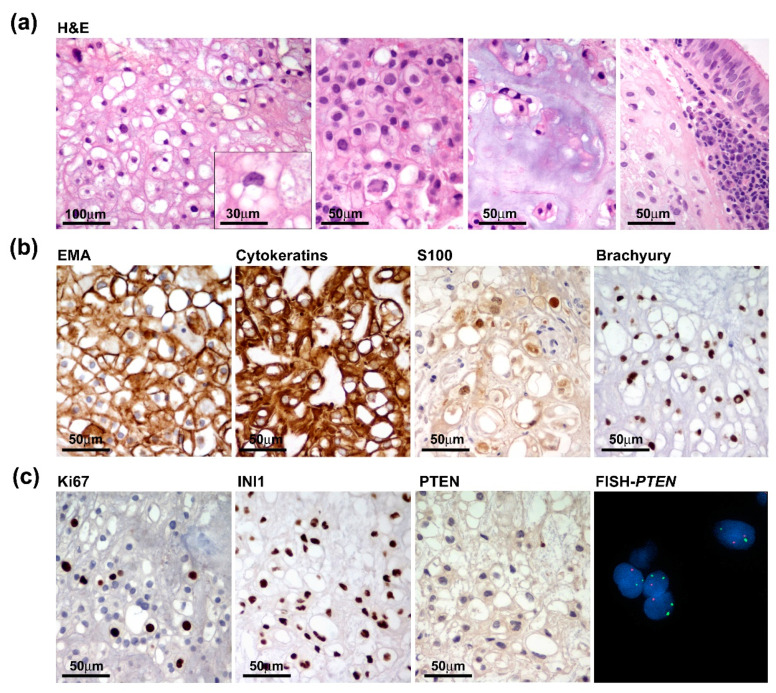
Pathological features of the primary tumour. (**a**) Tumour was composed of nests of large epithelioid cells with light eosinophilic (**left** two panels) or bubbly vacuolated cytoplasm (physaliphorous cells) (inset) with focal chondroid differentiation and evidence of sphenoidal bone infiltration with extension to the adjacent mucosa (**right** panels). (**b**) Tumour cells diffusely express chordoma markers such as EMA, Cytokeratins, S100 and Brachyury. (**c**) Proliferation index, assayed by Ki67/Mib1 immunostaining, only focally reached 7–8% and INI1 expression was retained (**left** panels), while PTEN was negative and hardly detected consistent with *PTEN* heterozygous deletion as assayed by FISH analysis (**right** panels).

**Figure 4 jpm-13-00425-f004:**
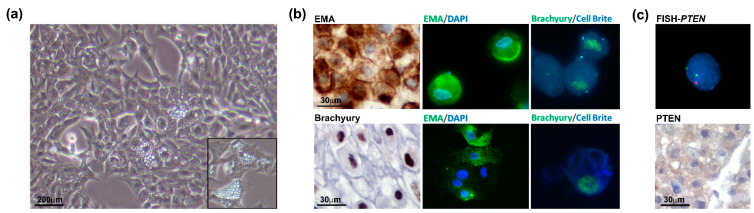
Chordoma cell line characterization. (**a**) Physaliphorous morphology of the chordoma cell line. The representative images of the CH3 cell line were taken at a magnification of 10× by a bright-field microscope. (**b**) CH3 cells express chordoma markers EMA and Brachyury during passages (P20, IHC, left panels; P10 IF, upper panels; P33, IF lower panels). For IF images, immunostaining of cells was performed for EMA and Brachyury, nuclei were stained using DAPI dye and cytoplasmic membranes using Cell Brite Blue. IHC images were taken at 60× and IF images at 63× original magnification (with immersion). (**c**) *PTEN* heterozygosity was confirmed in the CH3 cell line. The analysis was performed by FISH using PTEN/CEP10 probe, and nuclei were stained using DAPI (**top** panel). Immunohistochemistry on the cell block showed a barely detected expression of PTEN protein (**lower** panel).

**Figure 5 jpm-13-00425-f005:**
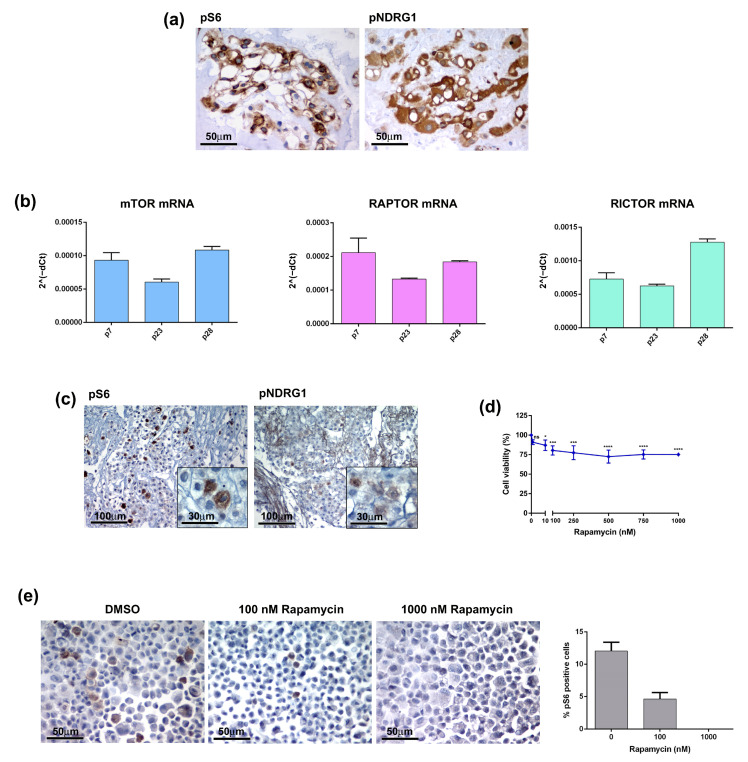
The mTOR pathway in chordoma tumour and cell line. (**a**) The mTOR pathway is activated in patient-derived chordoma tissue. The expression of mTOR markers, pS6 and pNDRG1, was analysed by IHC on FFPE tissue. (**b**) The CH3 cell line expresses mRNA of mTOR components. The mRNA expression of mTOR, Raptor and Rictor was analysed by quantitative real-time PCR. Bars and error bars represent mean values and standard deviation, respectively. (**c**) The markers of the mTOR pathway are expressed in the chordoma cell line. The expression of pS6 and pNDRG1 was analysed by IHC on FFPE cell blocks. (**d**) The mTOR inhibitor affected the CH3 cell viability. Cells were treated with 0–1000 nM Rapamycin for 10 days, the cell viability was analysed by MTT. Points and error bars on the curve represent mean values and standard deviation, respectively. ns, non-significant, * *p* < 0.05, *** *p* < 0.001, **** *p* < 0.0001. (**e**) Rapamycin affected the pS6-positive cells. Cells were treated with Rapamycin for 10 days, the pS6 expression was analysed by IHC and pS6 positive cells were quantified by ImageJ software.

## Data Availability

All data generated or analysed during this study are included in this published article.

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
