# Peer review of "Targeting mTOR Pathway in PTEN Deleted Newly Isolated Chordoma Cell Line"

_jpm, 2023, doi:10.3390/jpm13030425_

Round 1

Reviewer 1 Report

This is an interesting article about a new line of chordoma tumor that can be treated by Rapamycin. The authors explain widely the histological treatment of the samples to obtain enough data to present a rarely kind of chordoma with PTEN deletion. Chordoma is a rare tumor with a very difficult treatment options. I congratulate the authors for the effort to show a new possible treatment for this challenging tumors.

Author Response

We thank the Reviewer for the critical and careful review of our manuscript and for the overall positive impressions. The manuscript has been revised according to the reviewers’ suggestions as reported in the following point-by-point response. Changes have been made within the main text (marked up in yellow) and the revised text has been uploaded.

Reviewer: English language and style are fine/minor spell check required.

Answer: As suggested, the manuscript has been carefully reviewed and revised.

Reviewer: This is an interesting article about a new line of chordoma tumor that can be treated by Rapamycin. The authors explain widely the histological treatment of the samples to obtain enough data to present a rarely kind of chordoma with PTEN deletion. Chordoma is a rare tumor with a very difficult treatment options. I congratulate the authors for the effort to show a new possible treatment for this challenging tumors.

Answer: We thank the Referee for the kind comments and the positive impression regarding our manuscript.

Reviewer 2 Report

The paper is very well written and the results are well documented. However, have a few questions/observations for the authors:

1. is chondroma definitely a malignant neoplasm (as written in the abstract) in the histological sense?

2. please give the full name of the abbreviation where first used, e.g. TBXT, PTEN, mTOR

3. kindly state the purpose of the study at the end of the Introduction paragraph

4. is the study based on cellular material collected from 1 person?

5. please consider discussing the results more widely

Author Response

We thank the Reviewer for the critical and careful review of our manuscript and for the overall positive impressions. The manuscript has been revised according to the reviewers’ suggestions as reported in the following point-by-point response. Changes have been made within the main text (marked up in yellow) and the revised text has been uploaded.

Reviewer: The paper is very well written and the results are well documented. However, have a few questions/observations for the authors:

Answer: We thank the Referee for the careful revision and the positive impression regarding our manuscript.

Reviewer: 1. is chondroma definitely a malignant neoplasm (as written in the abstract) in the histological sense?

Answer: We thank the reviewer for this interesting observation. Indeed, as described within the most recent WHO classification of Central Nervous System Tumours (International Agency for Research on Cancer; 2021, 5th edition) and WHO classification of Soft Tissue and Bone Tumours (International Agency for Research on Cancer; 2020, 5th edition) “Chordomas are a family of primary malignant bone neoplasms demonstrating notochordal differentiation”. Staging is according to bone sarcoma protocols and outcome strictly depends on histological subtypes (comprising conventional, chondroid, poorly differentiated and dedifferentiated subtypes) and molecular alterations. Usually is locally invasive, but the most aggressive subtypes (e.g. dedifferentiated and IN1 deleted) may also show metastasis to lung, bone, lymph nodes and subcutaneous tissue.

Reviewer: 2. please give the full name of the abbreviation where first used, e.g. TBXT, PTEN, mTOR

Answer: Full names of the abbreviations have been provided within the revised text.

Reviewer: 3. kindly state the purpose of the study at the end of the Introduction paragraph

Answer: As suggested, the manuscript has been carefully reviewed and revised.

Reviewer: 4. is the study based on cellular material collected from 1 person?

Answer: Yes, and it has been clarified within the revised text.

Reviewer: 5. please consider discussing the results more widely

Answer: As suggested, the manuscript has been carefully reviewed and revised.
